# Effect of a Whey Protein Supplement on Preservation of Fat Free Mass in Overweight and Obese Individuals on an Energy Restricted Very Low Caloric Diet

**DOI:** 10.3390/nu10121918

**Published:** 2018-12-04

**Authors:** Anne Ellegaard Larsen, Bo Martin Bibby, Mette Hansen

**Affiliations:** 1Section of Sport Sciences, Department of Public Health, Aarhus University, Dalgas Avenue 4, 8000 Aarhus C, Denmark; 2Section of Biostatistics, Department of Public Health, Aarhus University, Bartholins Alle 2, 800 Aarhus C, Denmark; Bibby@ph.au.dk

**Keywords:** VLCD, body composition, weight loss, protein supplement, FFM

## Abstract

The obesity epidemic has caused a widespread interest in strategies to achieve a healthy “high quality” weight loss, where excess fat is lost, while fat free mass (FFM) is preserved. In this study, we aimed to examine the effect of whey protein supplementation given before night sleep on FFM preservation during a 4-week (wk) period on a very low caloric diet (VLCD). Twenty-nine obese subjects (body mass index (BMI) > 28 kg/m^2^) completed a 4-week intervention including a VLCD and a walking program (30 min walking × 5 times per week). Subjects were randomly assigned to either control (CON, n = 15) or a whey protein supplement (PRO, 0.4 g protein/kg/day, n = 14), ingested before bedtime. Body composition (dual-energy X-ray absorptiometry, DXA), blood analysis and physical test were performed pre and post intervention. We measured nitrogen excretion in three 24 h urine collections (Day 0, 7 and 28) to assess nitrogen balance. Changes in nitrogen balance (NB) after 7 and 28 days was different between treatment groups (interaction *p* < 0.05). PRO was in NB after 7 days and in positive NB at day 28. In contrast, CON was in negative NB at day 7, but in NB at day 28. Nevertheless, no significant group differences were observed in the change in pre- and post-FFM measurements (−2.5 kg, [95% CI: 1.9; 3.1], *p* = 0.65). In conclusion, ingestion of a whey protein supplement before bedtime during a 4-week period on a VLCD improved nitrogen balance, but did not lead to any significant improvement in the quality of the weight loss in regard to observed changes in body composition and health parameters compared with controls.

## 1. Introduction 

Very low caloric diets (VLCD) used for four weeks or greater are effective in regards to inducing weight loss and improvements in risk factors related to metabolic syndrome. Nevertheless, weight loss induced by VLCD is associated with a significant reduction of fat free mass (FFM). Eston et al., (1992) observed that loss of FFM accounted for 37% of the total weight loss after six weeks on a VLCD [1]. Preserving or minimizing loss of FFM while losing fat mass (FM) is optimal and has been referred to as a “high-quality” weight loss [2], since FFM is closely related to basal metabolic rate, functional capacity and lifespan [3,4,5,6]. 

Regular physical training attenuates the loss of FFM during weight loss [7]. In addition, the protein content of energy-restricted diets seems to be crucial when aiming at minimizing FFM loss during weight loss [2,8]. Pasiakos et al. (2013) reported that a daily intake twice the Recommended Dietary Allowance (RDA) of protein was associated with FFM preservation in physically active men on a calorie restricted diet (40% energy deficit) [9]. 

The timing of protein supplementation may be of importance for the benefit of the supplement on FFM preservation in the catabolic state [2]. Adding 0.4 g protein/kg whey protein on top of main meals may not be as effective as ingesting the protein supplement between meals [10]. In contrast, an extra protein supplement (compared to no supplement) ingested before bedtime improves protein balance and muscle protein synthesis [11]. Furthermore, protein ingestion before bedtime seems to enhance muscle growth in response to regular resistance training on FFM in young healthy men [12]. However, elucidation is needed of the effect of combining an exercise program and a protein supplement on FFM loss during severe caloric restriction (VLCD) in obese subjects.

In the present study, we investigated effects of a whey protein supplement before bedtime in addition to a VLCD and a walking program on changes in nitrogen balance, body composition and metabolic risk factors in individuals with overweight and obese compared to no additional protein. We hypothesized that an extra protein supplement before bedtime would reduce the loss of FFM and thereby lead to a “high-quality” weight loss compared to controls with no additional protein supplement. The control group was instructed to follow typical VLCD guidelines with no additional energy intake. 

## 2. Materials and Methods

### 2.1. Design

We conducted a randomized, single-blinded, controlled four-week intervention trial, where all included individuals were overweight or obese and followed a VLCD and walking exercise program. We performed stratified randomization of the included women and men after we had allocated them by sequentially numbered containers to the Control group (CON) or the Protein group (PRO). PRO consumed a protein supplement before bedtime each night, with no added supplement for CON. Before and after the intervention period the subjects went through an experimental test day. The duration from enrollment to the final experimental day was 5 to 6 weeks. The VLCD powder (Nutrilett, Orkla Health A/S, Oslo, Norway) and protein supplement (Maxim, Orkla Health A/S, Oslo, Norway) was handed out by a person not involved in the collection of data, data analysis or the the random allocation of subjects. The physical tests and the dual-energy X-ray absorptiometry (DXA) scans were performed by test personal blinded to the group allocation. We enrolled participants based on the inclusion and exclusion criteria. Figure 1 shows an overview of the study design and experiment test days. Additionally, each subject signed a weekly questionnaire of compliance to the diet and supplement. 

### 2.2. Subjects

Individuals who were overweight (body mass index (BMI) > 28 kg/m^2^) and aged between 18 and 55 years were recruited via posters, newspaper, social media, and through referral from the municipality’s health center in the period from 7 January until 21 February 2016. Exclusion criteria were participation in regular resistance-, aerobic- or anaerobic training (>1 h/week within the last 3 months), diabetic type I, and physical disabilities that would hinder completion of the walking program. Forty-one subjects met the criteria for participation and were randomly assigned to either VLCD + walking exercise training (CON: n = 21), or VLCD + walking exercise training + whey protein concentrate (WPC) supplement (PRO: n = 20) (Figure 2). The study included healthy women and men who were overweight in the age range 21 to 55 years (average 41 years). 

All subjects gave their informed consent to participate prior to the intervention. We carried out the trial in accordance with the Declaration of Helsinki, approved by the local ethics committees (journal nr.1-10-72-344-15) and registered at Clinical.Trials.gov (NCT03102372). The screening and the experimental days were all performed at Section for Sports Science, Department for Public Health, Aarhus University.

### 2.3. Diet

All subjects were on VLCD (~690 kcal/day, Nutrilett, Orkla Health A/S, Norway) for four weeks. The Danish Veterinary and Food Administration approve the VLCD products used in the present trial. We provided the subjects with a VLCD divided in 6 portions per day as either shakes (112 kcal/portion) and/or soups (122 kcal/portion) (Table 1). We allowed the subjects to consume non-caloric beverages and sugar-free chewing gum. 

The PRO group ingested a whey protein isolate supplement just before bedtime, which contained 0.4 g protein/kg (Maxim, Orkla Health A/S, Oslo, Norway) (Table 1). The subjects and the laboratory staff who performed the physical tests and DXA scans were blinded for the content of the supplement. The addition of a protein supplement (thereby addition of energy) to a standardized VLCD diet was chosen to optimize the quality of the weight loss by preventing FFM loss. 

### 2.4. Exercise Training

All subjects followed a walking program during the intervention (30 min of walking × 5 times per week). The subjects were recording walking time and distance in the mobile app Endomondo (Under Armour Connected Fitness, Austin, TX 78701, USA). We instructed the subjects not to alter their general daily physical activity level besides the walking program and they recorded whether they were primarily sedentary or active during general working hours in a standardized questionnaire. A greater volume of aerobic and resistance training seems to be optimal for achieving fat loss and preserving FFM during energy restriction [13,14]. Nevertheless, we only included a walking program as part of the intervention since we were aiming at fulfilling the general recommendations for physical activity and were not aiming at elucidating the maximal muscle perceiving effect of physical training during energy restriction.

### 2.5. Experimental Days before and after the Intervention Period

We performed the individual test of the subjects at the same time of the day on the two experimental days. Anthropometrics were assessed using standardized procedures. Height was initially measured by a stadiometer (±1 cm) and body weight (±100 g) was measured using a *Tanita SC 330* (Tanita Europe B.V, 2132 NG Hoffddorp, the Netherlands). Body composition was measured using dual-energy X-ray absorptiometry (DXA) (GE Lunar DXA scan, GE Healthcare, Chicago, IL, USA) and determined using the system’s software package (enCore version 17 software GE Healthcare). Waist and hip circumferences were assessed by a measuring tape (±0.5 cm) using standard procedures [15].

We collected three 24 h urine samples at day 0, 7 and 28 of the intervention period (Figure 1). Urine volume (g) was measured and a ~10 mL urine sample from each collection period was stored at −20 °C until further analysis for urea (*NPU03930*) at Aarhus University Hospital, Aarhus, Denmark.

Nitrogen intake was calculated by dividing daily protein intake by 6.25 g [8]. Nitrogen (N) excretion was calculated based on the measured g (mass) of urine and the urea concentration using the formula: N excretion in urine (g/day) = (urine g × urea concentration) × molar mass of urea (60.06 g/mol) × N content (46.64%). Total nitrogen excretion was calculated by adding 2.5 g/day of nitrogen, representing non-measurable nitrogen loss [16]. Nitrogen balance (NB) was calculated using the formula: NB (g/day) = nitrogen intake (g/day) ÷ nitrogen excretion (g/day) [8]. 

All blood samples we obtained after overnight fasting. We obtained blood by finger sticks and immediately analyzed for capillary blood (cb)-glucose (*Hemocue Glucose 201 RT* [17,18]), total cb-cholesterol (*Accutend plus* [19,20]) and cb-ketones (*Freestyle Precision Neo* [21]). In addition, we collected venous blood, which we centrifuged at 22 °C for 15 min at 1200× *g* and stored at −20 °C until further analysis for serum (s)-insulin (*NPU02497*) and s-free triiodothyronine (s-T3) (*NPU03625*). In addition, we analyzed the blood for s-testosterone in men (*NPU03549*) and s-estradiol in women (*NPU09357*). Aarhus University Hospital, Denmark performed the analyses. Blood pressure (SBP; systolic blood pressure, DBP; diastolic blood pressure) was measured three times in a seated position. After tests the subjects received a standardized meal; a banana and a meal replacement bar (*Nutrilett, Orkla Health, Oslo, Norway*) containing 220 kcal (14 g protein, 22 g carbohydrate and 7 g fat).

We evaluated aerobic fitness level using an Aastrand two-point bike test [22,23]. The subjects performed the Aastrand two-point bike test on an electronically braked cycle ergometer (*Monark 928 E, Monark Exercise AB, Vansbro, Sweden)* after four min standardized warm-up (75 W, cadence 55 rpm). The 10 min test period was divided in two periods. In the first 6 min period, the resistance was ~100 W (cadence at 80 rpm) and during the following 4 min, the resistance was increased by at least 60 W (80 rpm). The resistance during the test was individually adjusted so the heart rate reached ~100 bpm during the first part of the test and at least 20 bpm higher during the last minute of the test. The heart rate was monitored (*Polar electro N2965, Polar Electro, Kempele, Finland*) every 15 s in the last min of each period. The workload (W) was similar during the test before and after the intervention. To enhance performance, the test personnel used standardized verbal encouragement and motivational chants.

### 2.6. Statistical Analysis

Prior to intervention, we performed a sample size calculation. Earlier studies using VLCD have reported 3.7–7.8% decline in fat mass (FM) after 6 to 16 weeks of intervention [7,14,24]. We assumed that PRO would maintain FFM (in contrast to CON) [25], but experience a greater decline in FM (SD = 3%). Level of significance and power were determined to be 5% (A = 1.96) and 80% (B = 0.84). Sample size required to detect the hypothesized difference in body fat percentage was calculated to be 16 subjects within each group based on the following formula [A + B]^2^ × 2 × (SD/DIFF)^2^.

Between groups, differences in baseline characteristics were evaluated using unpaired Student’s *t* tests. For variables measured before and after the intervention period, a two-way mixed ANOVA with repeated measures was performed to test for changes due to group, time and interaction between group and time. In the two-way mixed ANOVA, we also tried to adjust for different individual characteristics, such as activity level at job, diet and initial weight. All statistical analyses were performed using Stata 12. GraphPad Prism 7.0a was used to design the figures. Data are presented as mean 95% CI, unless otherwise described. *p* < 0.05 was considered statistically significant. 

## 3. Results

Twelve subjects were lost to follow up because of non-compliance with the diet (*n* = 6), constipation (n = 3) and influenza (*n* = 3), whereas 29 subjects completed the intervention (PRO; *n* = 14 and CON; *n* = 15) (Figure 2). The daily intake of protein was in total 70 g/day in CON and in average 111 g/day in PRO, whereas the total daily energy intake was 690 kcal/day in CON and in average 901 kcal/day in PRO (Table 1). Table 2 and Table 3 show subject characteristics for completers. At baseline, no significant difference between groups was observed in body weight, BMI, FFM, FM, android- and gynoid FM, waist circumference, hip circumference, nitrogen excretion, blood pressure, cb-cholesterol, cb-ketones, s-testosterone, s-estradiol, s-T3, s-insulin, fitness and estimated maximal oxygen uptake (Table 2 and Table 3). Adherence to VLCD was good when evaluating questionnaire reports (CON 99.4 ± 0.7% and PRO 99.2 ± 0.6% out of the 42 portions). The significant weight loss supported the self-reported adherence to the diet and compliance with the diet. Ingestion of a protein beverage just before sleep did not influence sleeping hours and sleeping quality when evaluating self-reports. 

### 3.1. Body Composition

Body weight was reduced (*p* < 0.001) by 8.6% in CON and 7.8% in PRO (Figure 3), but the change in absolute and relative weight loss in kg (Table 2) did not differ between groups (Table 2). 

Reduction in FFM was significant in both groups (*p* < 0.001), but the reduction did not differ between groups (interaction *p* = 0.65, Table 2). However, a higher initial body fat percentage was associated with an improved preservation of FFM in legs (r^2^ = 0.33, *p* = 0.00), but a greater loss of FFM in the torso (r^2^ = 0.15, *p* = 0.04). In contrast, initial weight per se showed no correlation with changes in body composition.

Similar to the results for absolute FFM loss, the relative reduction in FFM compared to total weight loss did not differ between groups (interaction, *p* = 0.29, Figure 3). The relative FFM loss corresponded to 26.4% in CON and 31.2% in PRO. 

The absolute loss of FM was significant (CON; 6.3 [95% CI: 5.68; 6.94] kg and PRO; 5.5 [95% CI: 4.61; 6.36], but did not differ between groups (interaction, *p* = 0.11).

We observed a significant reduction in android fat and gynoid fat mass (measured by DXA) and waist and hip circumferences in both groups, but the size of the reduction did not differ significantly between groups (Table 2). The reduction of android fat was significantly greater than reduction in gynoid fat in both groups. In support, we observed a similarly greater absolute reduction in waist circumference than hip circumference. 

### 3.2. Nitrogen Balance

Figure 4 shows data for nitrogen intake g/day, nitrogen excretion g/day, and nitrogen balance g/day 4. After 7 days of intervention, the NB was different between groups (CON: −1.5 [95% CI: −2.01; −1.29] g/day; PRO: 1.3 [95% CI: −0.52; 3.06] g/day, *p* = 0.01). Similarly, at day 28 NB was greater in PRO (3.2 [95% CI: 1.03; 5.28] g/day) than CON (−0.48 [95% CI: −2.38; 1.42] g/day, *p* = 0.01). Comparing these data to a NB at zero PRO demonstrated an equilibrium NB at day 7 (*p* = 0.15) with a significant change to a positive NB (*p* = 0.006) at day 28. In contrast, CON demonstrated a significant negative NB (*p* = 0.03) at day 7 returning to an equilibrium NB at day 28 (*p* = 0.59) (Figure 4C). 

### 3.3. Blood Parameters

Nine subjects (CON *n* = 6, PRO *n* = 3) completed the last experimental day two to five days after the last intervention day instead of on the day immediately following the intervention. Excluding these nine subjects did not change the conclusion for any of the parameters and data from all subjects are included in the presented data (Table 3).

Fasting cb-glucose at baseline was within normal range for all subjects (≤5.6 mmol/L [26]). The results after 4 weeks of intervention showed a significant reduction in fasting cb-glucose in CON, but not in PRO (Table 3). Fasting cb-ketone bodies were significantly increased after 4 weeks (CON; 0.3[95% CI: 0.04; 0.57] mM; PRO; 0.5 [95% CI: 0.21; 0.81] mM), but no significant interaction was observed. Total cb-cholesterol was at baseline above >5 mmol/L in 6/15 subjects in CON and 6/14 subjects in PRO. After the intervention this number was reduced to 4/15 in CON and 0/14 in PRO, with no significant interaction observed in absolute cb- cholesterol values (*p* = 0.08). 

Testosterone in the men was higher after the intervention (*p* = 0.01), but the increase did not differ between groups (interaction, *p* = 0.32). We observed no change in estradiol in the women after the intervention. S-T3 was within normal range (3.0- 6.5 pmol/L [27]) within both groups at baseline. A significant reduction in s-T3 was observed after the intervention in both CON (−7%) and PRO (−15%) (interaction, *p* = 0.60). S-insulin in both groups was above normal values (18–90 pmol/L [28]) at baseline, but was significantly reduced after the intervention (*p* < 0.01) to a similar extent in both groups (interaction, *p* = 0.18).

### 3.4. Blood Pressure

Blood pressure was higher than recommended (SBP < 120 mmHg and DBP < 80 mmHg [29]) at baseline in 12/15 in CON, and 9/14 subjects in PRO (Table 3). These numbers were reduced to 5/15 in CON and 3/14 in PRO subjects (Table 3). After the intervention blood pressure was significantly lowered (*p* < 0.05) in both groups with no significant difference observed between groups (SBP; interaction *p* = 0.44 and DBP; interaction *p* = 0.08). 

### 3.5. Aerobic Capacity

Walking time and walking distance did not differ between groups. A greater portion of subjects in CON (60%) categorized themselves as active at work compared to PRO (36%, *p* < 0.001). 

Results from the Aastrand two-point bike test showed no significant change in estimated maximal oxygen uptake (CON: −0.2 [95% CI: −0.48; 0.10] O_2_ mL min^−1^; PRO: 0.07 [95% CI: −0.14; 0.28] O_2_ mL min^−1^, interaction *p* = 0.12). However, a significant but similar increase in fitness level was observed in both groups (CON: 3.1 [95% CI: 0.52; 5.69] O_2_ mL min^−1^ kg^−1^; PRO: 5.3 [95% CI: 2.09; 8.47] O_2_ mL min^−1^, interaction *p* = 0.26). 

## 4. Discussion

The main novel finding in the present trial was that consuming an extra bolus of whey protein before bedtime while undertaking a 4-week VLCD did not improve preservation of FFM compared to ingestion of VLCD alone. However, both groups experienced comparable optimization of body composition and reductions in metabolic risk factors.

### 4.1. Body Weight and Fat Mass

On average, the subjects’ reductions in body weight and fat mass were comparable with previous VLCD studies. The energy intake in PRO was 30.6% higher than in CON due to the protein supplementation, which corresponded to ~5824 kcal in total during the intervention. Assuming an energy deficit of 1 MJ corresponds to a ~240 g weight loss [16] this difference should have led to a ~1.4 kg greater weight loss in CON. Furthermore, CON reported a higher physical activity level at work than PRO, which would also support an increased energy deficit in CON and improve the weight loss. However, the weight loss was only 0.6 kg greater in CON and not significantly different from PRO. There may be several explanations for this observation. Previously, a protein-rich diet results in a greater 24h energy expenditure than an isocaloric carbohydrate-rich diet [30]. Nevertheless, the reduction in s-T3, which is a well-documented response to weight loss [16,31,32], was not significant different between groups, which suggests a comparable lowering of metabolic rate [33] within both groups. Another explanation could be a higher general activity level in PRO. Use of accelerometers or double-labelled water to document this would have been optimal. Thirdly, we cannot rule out that PRO have improved their weight loss due to water excretion which, during a hypocaloric protein-rich diet, has been reported to be greater than during a hypocaloric carbohydrate-rich diet [34]. 

### 4.2. Effect of Protein Supplement on Nitrogen Balance and Fat Free Mass

Protein supplementation provides building blocks for gluconeogenesis, which has shown to reduce breakdown of endogenous protein sources in an energy deficit state [9,25,35]. Furthermore, protein supplementation before sleep enhances muscle protein synthesis and thereby protein balance during the night [12,36]. In support of a positive effect of protein supplementation, we estimated NB in PRO to be positive after 28 days on VLCD in contrast to neutral in CON. A positive effect of an enhanced protein content on NB during a hypocaloric diet is supported by others [37]. Therefore, our NB data indicates a beneficial effect of protein supplementation on FFM preservation compared to CON. However, we observed no significant difference in relative (31.2% vs. 26.4% of lost weight in PRO and CON) and absolute loss of FFM (2.7kg vs. 2.4 kg of lost FFM in PRO and CON). The relative loss of FFM was in accordance with previous VLCD studies not including extra protein supplements [1,38,39]. A prolonged intervention period might have resulted in a significant improvement in FFM preservation in PRO compared to CON.

The effect of adding a protein supplement on the top of a VLCD to reduce loss of FFM has not previously been reported. However, during less restrictive hypocaloric diet (>901 kcal/day) enhancing protein intake (>0.8 g/kg/day) enhances FFM preservation [2,9,37,40,41,42,43]. Nevertheless, no effect of additional protein during weight loss has also been reported [25,44], which is in line with the present findings. The conflicting results may be due to the size of the energy deficit or various other factors (the type and volume of physical activity during the intervention, the amount or source or timing of the additional protein [2,11,45], etc.). 

In the present trial, we administrated the protein dose relative to body weight (0.4 g/kg/day). The PRO group ingested in average 1.08 g protein/kg/day in total compared to 0.69 g protein/kg/day in the CON group. Based on results from a meta-analysis, Krieger et al. (2006) recommended that in an energy deficient state the daily protein intake should exceed 1.05 g protein/kg/day to improve FFM preservation [46]. Other findings support this [2,9,37,40,41]. Therefore, we cannot exclude that enhancing the amount of protein in the supplement would have led to preservation of FFM since the energy deficit in the present trial was greater than in the meta-analysis by Krieger et al. [46]. We wanted to induce a marked weight loss in PRO and therefore we did not want to enhance the total energy intake more by adding more protein to the supplement in PRO.

### 4.3. Effect of Walking Intervention on Fat Free Mass

We instructed the subjects to perform a walking intervention to standardize activity level and minimize possible impact of different training load. Resistance-based exercise was not performed, in order to avoid impact on FFM changes, with a protein supplement as the primary intervention. The literature supports that adding exercise to an energy-restricted diet reduces the relative FFM loss [38,47]. In a meta-analysis, Ballor and Poehlman (1994) observed that adding moderate intensity exercise to an energy-restricted diet program reduced FFM loss from 28% to 13% in men and 24% to 11% in women [47]. Nevertheless, our results indicate that the intensity or volume of the walking program in the present trial was insufficient to induce FFM preservation since the relative loss of FFM was comparable with the loss in the control groups in Ballor and Poehlman [47]. However, subjects with the greatest initial body fat percentage experienced an improved preservation of FFM in legs (r^2^ = 0.33, *p* = 0.00), which may be because the load on the leg muscle during walking was relatively higher in these subjects compared to subjects with a lower body fat percentage. In fact, the two subjects with the highest initial fat percent gained leg FFM (0.3 and 1.1 kg). Both groups followed the same walking program, but a higher activity level at work was reported in CON (categorized as active at work; CON 60% vs. PRO 36%, *p* < 0.05). This could be a negative confounder in regards to observing FFM preservation in response to protein supplementation. Nevertheless, adjusting for the reported activity level at work did not change the conclusion, showing no significant group difference in FFM preservation. 

### 4.4. Effect of the Intervention on Fitness and Health Parameters

VO_2_max did not change, but due to the weight loss, the fitness level (maximal mL O_2_/min/kg) was higher after the intervention period. In contrast, others have reported improvements in VO_2_max when combining regular aerobic exercise with a hypocaloric diet [7,48,49]. The discrepancy is probably explained by the subjects in our study exercising at relatively low intensity. In addition, we measured the physical performance just after intervention ended when the subjects were still in an energy deficit state. Therefore, the availability of glucose and glycogen as energy fuel was probably lower during the test at the end of the intervention compared to baseline. In support, at the experimental day after the intervention the concentration of insulin was lower and the concentration of ketone bodies higher. This indicates that during the indirect submaximal bike test, fat oxidation was probably higher during the post testing at the similar workload than when testing at baseline. Since oxygen cost/adenosine triphosphate (ATP) production is higher when using fat as fuel compared to glucose, this may have induced an increase in heart rate and thereby confounded the estimation of VO_2_max. 

Lipid profile and blood pressure were either unchanged or improved after 4 weeks of intervention, which indicates a reduced risk of cardiovascular disease and metabolic abnormalities. This observation is in agreement with previous weight loss interventions [50,51,52,53]. The improvements the subjects experienced did not differ between groups. This is in line with the changes in body composition and fitness level, which were comparable between groups. 

We measured sex hormones to investigate whether diet or/and changes in body composition would be related to changes in sex hormones. Testosterone (only measured in males) was relatively low at baseline, but a significant increase was observed post intervention, which is supported by previous weight loss trials [54]. This observation may be related to BMI which is inversely proportional to circulating testosterone due to testosterone being converted to estradiol in adipose tissue.

### 4.5. Limitations

The energy intake was not energy matched since we added the protein supplement on top of VLCD. Therefore, we cannot determine whether our findings are due to the higher energy intake in PRO or the whey protein supplement. However, we were aware of this when designing the project. The VLCD we used was a commercial standardized product that meets the requirement and standards for VLCD. We did not want to give the CON group an isocaloric carbohydrate supplement on top of this standardized product, since in the real world that would not be the practice unless the subjects are not sticking to the diet. The perspective was that we wanted to optimize the quality of weight loss by testing the effect of a new approach to optimize the VLCD formula. Another reason for not adding an isocaloric sugar/carbohydrate supplement to the control was that the insulin response would potentially influence the primary outcome parameter, the FFM loss, by reducing the protein breakdown rate. 

### 4.6. Future Directions

We chose to use whey protein supplement in the present study because it is protein source of high quality [55,56]. Whey protein has a high content of essential amino acids to stimulate muscle protein synthesis and a specific high content of leucine compared to other protein sources. Whey protein isolate effectively stimulates skeletal muscle synthesis [57]. Furthermore, whey protein triggers a higher secretion of insulin than a slowly absorbed protein source such as casein, which might be beneficial since insulin reduces muscle protein breakdown [58]. Nevertheless, a slow release protein source such as casein could potentially be more beneficial than a whey protein supplement in an energy-restricted situation since a rapid increase in plasma amino acid concentration as observed after whey protein ingestion leads to a higher oxidation of the amino acids [57]. Therefore, we cannot say whether it would have changed our results if we had supplemented with casein instead of a whey protein supplement. Future studies should aim for elucidating whether a slowly absorbed protein supplement can positively optimize weight loss quality during VLCD by reducing the loss of FFM.

### 4.7. Conclusions

During a VLCD period combined with a walking program, a whey protein supplement before bedtime did not lead to preservation of FFM or changes in metabolic health parameters compared with a VLCD and walking alone. The additional calorie intake in the PRO group did not reduce the size of the weight loss compared to the control group. The benefit of physical activity on fat free mass maintenance seems positively related to initial body fat percentage and working load.

## Figures and Tables

**Figure 1 nutrients-10-01918-f001:**
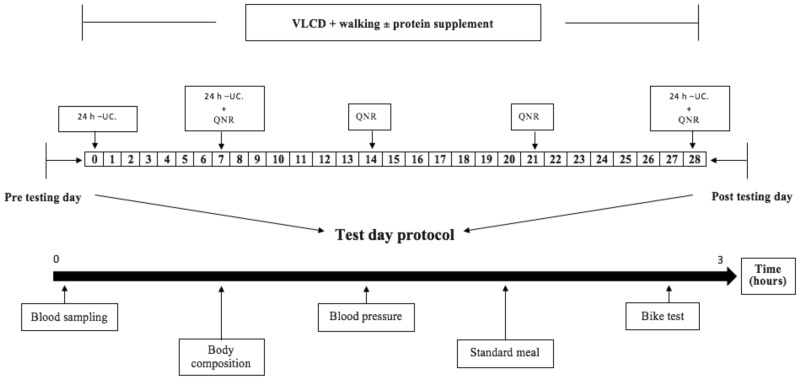
Overview of the design. The pre- and post-tests were performed maximally five days after the intervention. 24h-UC: 24-h urine collection. QNR: questionnaires. VLCD: very low caloric diet. Body composition: dual-energy X-ray absorptiometry (DXA) scan and hip and waist circumference measurements. Bike test: Aastrand 2–point bike test.

**Figure 2 nutrients-10-01918-f002:**
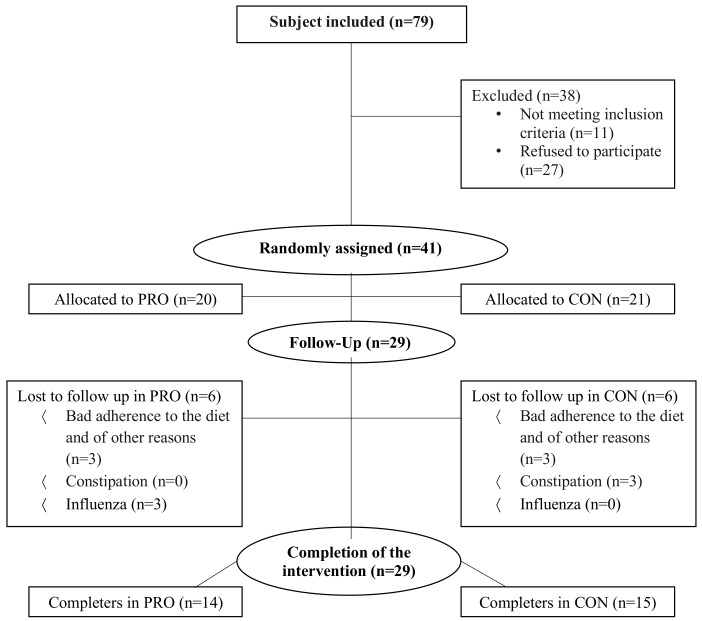
Flowchart of recruitment and completion of the trial. PRO: protein group, CON: Control group. n: number of subjects, FM: Fat mass, FFM: Fat free mass.

**Figure 3 nutrients-10-01918-f003:**
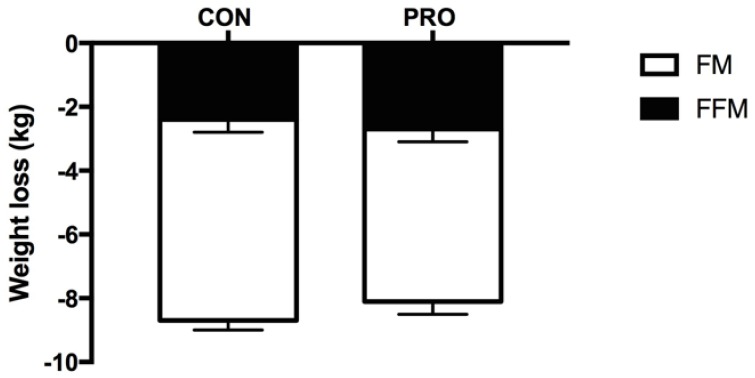
Body composition changes from baseline to the end of the intervention after 4 weeks. Data presented as mean ± SE.

**Figure 4 nutrients-10-01918-f004:**
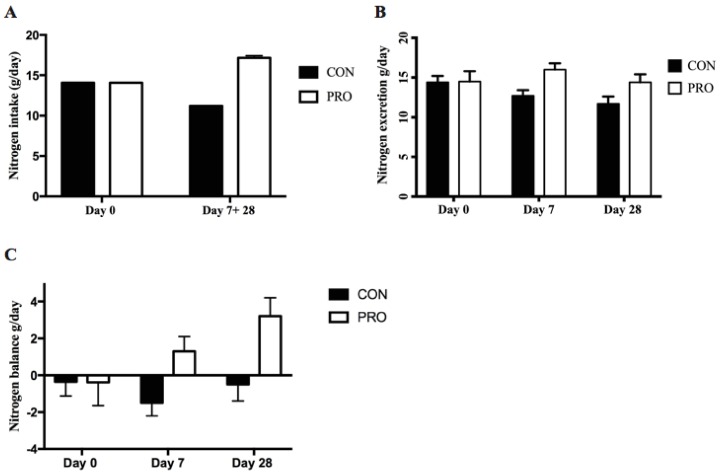
Nitrogen intake g/day (**A**) Nitrogen excretion g/day (**B**) and Nitrogen balance g/day (**C**). The results are calculated based on analysis of 24-hr urine collections and protein intake during the intervention. Data presented as mean ± SE.

**Table 1 nutrients-10-01918-t001:** Very low caloric diet (VLCD) and protein supplement.

	CON	PRO			
Nutrient Content	VLCD	Protein Supplement		VLCD + Supplement	
**Energy (kJ/kcal)**	2906/690	895/211		3801/902	
**Protein (g/day)**	70	41.3	[30.8; 52.4]	111.3	[100.8; 122.4]
**Fat (g/day)**	15	3.3	[2.5; 4.2]	18.3	[17.5; 19.2]
**Carbohydrate (g/day)**	80	3.6	[2.7; 4.6]	83.6	[82.7; 84.6]
**Fiber (g)**	20.4	0.4	[0.2; 0.5]	20.8	[20.6; 20.9]

Note. kJ; kilojoule, kcal; kilocalorie, G; gram, CON; control, PRO; protein.

**Table 2 nutrients-10-01918-t002:** Subject characteristics.

	Control		Protein		Df Groups	Control		Protein		Interaction
**Weight (kg)**	102.1	[9.5; 10,9]	102.6	[92.7; 112.5]	0.92	−8.7	[7.8; 9.7]	−8.1	[6.7; 9.6] *	0.47
**BMI (kg/m^2)^**	35.1	[32.5; 38.3]	34.9	[32.2; 37.6]	0.99	−3.2	[2,8; 3.6]	−2.7	[2.4; 3.1] *	0.18
**FFM (kg)**	59.0	[52.0; 61.0]	57.2	[50.9; 63.5]	0.62	−2.4	[1.6; 3.2]	−2.7	[1.7; 3.6] *	0.65
**Body Fat (%)**	41.9	[37.5; 46.2]	44.1	[40.0; 48.1]	0.43	−3.0	[0.2; 0.4]	−2.2	[1.3; 3.0] *	0.35
**Android Fat (%)**	52.3	[47.8; 56.8]	54.6	[50.6; 58.5]	0.42	−4.6	[0.3; 0.6]	−3.4	[1.8; 5.1] *	0.26
**Gynoid Fat (%)**	41.6	[36.1; 47.1]	45.9	[40.7; 51.0]	0.23	−2.3	[1.5; 3.0]	−1.9	[0.1; −0.2] *	0.35
**Waist c (cm)**	111.0	[102.3; 119.3]	105.0	[97.9; 113.2]	0.89	−10.0	[8.6; 12.2]	−8.0	[5.9; 9.8]	0.05
**Hip c (cm)**	120.0	[113.6; 126.5]	121.0	114.1; 127.3]	0.89	−6.0	[3.6; 7.3]	−6.0	[4.8; 7.2] *	0.62

Note. Data are presented as mean [95% confidence interval]. * *p* < 0.001: significant differences between pre and posttest. A negative value indicates a reduction relative to the pre-test value. FFM, Fat free mass; Df groups; Differentiation between groups; BMI, Body Mass Index; C, Circumference. In the data analysis, we adjusted for several sociodemographic factors (age, sex and initial weight), but adjustment did not influence the results significantly. Therefore, Table 2 shows raw values.

**Table 3 nutrients-10-01918-t003:** Blood parameters and blood pressure.

	Control		Protein		Df Groups	Control		Protein		Interaction
**SBP (mmHg)**	139.0	[131.5; 146.9]	138.0	[125.1; 151.2]	0.88	−15.0	[8.3; 21.2] *	−18.0	[12.3; 23.4] *	0.44
**DBP (mmHg)**	87.0	80.7; 92.5]	88.0	[69.2; 80.2]	0.68	−8.0	[3.3; 12.3] #	−14.0	[8.5; 18.8] *	0.08
**Chol (mmol/L)**	5.1	[4.5; 5.6]	5.3	[4.6; 5.9]	0.66	−0.6	[0.1; 1.0] #	−1.1	[0.7; 1.6] *	0.08
**Ketones (mM)**	0.1	[0.1; 0.2]	0.1	[0.1; 0.2]	0.85	0.3	[0.0; 0.8] #	0.5	[0.2; 0.8] *	0.27
**Glucose (mmol/L)**	5.9	[5.4; 6.4]	5.4	[5.0; 5.9]	0.13	−0.7	[0.4; 1.1] #	0.1	[−0.8; 0.5]	0.02
**Testo (nmol/L)**	15.9	[10.2; 21.6]	11.1	[6.2; 16.0]	0.30	8.8	[1.5; 19.05] #	2.8	[0.9; 4.7] #	0.32
**Estradiol (pmol/L)**	232	[−232; 695]	176	[61; 291]	0.68	158	[−221; 536]	−9	[−170; 188]	0.77
**T3 (pmol/L)**	5.8	[5.3; 6.3]	5.4	[5.0; 5.8]	0.05	−0.6	[0.1; 1.1] *	−0.8	[0.4; 1.3] *	0.60
**Insulin (pmol/L)**	139.7	[73.8; 204.3]	118.0	[50.1; 186.0]	0.28	−75.4	[19.6; 72.6] *	−43.7	[−83.4; −4.1] #	0.18

Note. Data presented as mean [95% confidence interval]. * *p* < 0.001. # *p* < 0.05: showing significant differences between pre- and post-test. A negative value indicates a reduction relative to the pre-test value. All blood parameters we obtained in the morning in the post absorptive state. Df groups; differentiation between groups; SBP, systolic blood pressure; DBP, diastolic blood pressure, Chol, cholesterol; Testo, testosterone; T3, free triiodothyronine.

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
