# Peer review of "Effect of a Whey Protein Supplement on Preservation of Fat Free Mass in Overweight and Obese Individuals on an Energy Restricted Very Low Caloric Diet"

_nutrients, 2018, doi:10.3390/nu10121918_

Round 1

Reviewer 1 Report

In their manuscript, Larsen et al describe an intervention trial of protein supplementation during VLCD to prevent FFM loss. The study is well written and interesting and the authors do not find a significant preservation of FFM with protein supplementation. A few points should be addressed regarding clarity and experimental issues.

1)   The authors include all patients with BMI>28 kg/m2 in their study. Was there any attempt to see if differences were more pronounced when stratified by BMI?

2)   Do the authors think that the increase of >200 kcal/day with the protein supplement had a significant effect on the results of the study?

3)   Line 107: “Hunder” should be “Under”? Also please give a reference.

4)   Line 113: “realistic real world…” seems redundant

5)   Was any control made for time frame before “bedtime” the protein supplement was consumed or for what time the subject went to bed and number of hours slept?

6)   Were data controlled for age, sex and other sociodemographic factors in the calculations? Would be good to include this information in Table 2.

7)   The loss of subjects led to the number of individuals/group being below the statistical power calculation. Was any attempt made to recruit new subjects to ensure statistical power? Also, overall inclusion seems to be quite low. Please further comment on the exclusion and refusal to participate rates (which seem quite high).

8)   The loss of 3 subjects to influenza is an interesting point. Were these subjects vaccinated? It is also very interesting that all the subjects were in the PRO group. Did the authors follow up on this point at all?  

9)   The formatting on Table 2 makes it difficult to interpret. Please consider revising. Also, please consider bolding significant results.

10)                   The presentation of Figure 2 is difficult to interpret. Also, in the legend it ways changes “at baseline and after 4 week…”; however, it appears only the final weight loss is shown?

11)                   Please add a separate section and expand the discussion of possible sources of bias and confounding for this study.

Author Response

See the point- by point response in the uploaded Word file 

Reviewer 2 Report

The authors have completed a novel intervention that has been missing from the literature for some time. The work is extensive and overall has been conducted well. While some methodological aspects could have been improved (measured energy requirements before the study, measures of oxygen consumption during the submaximal/maximal testing, a resistance training intervention to maximise the use of the ingested protein), the study is still valid.

There are several comments requiring addressing to clarify the outcomes and the writing for the reader. Most these are the order of information in the text, or the incorrect descriptions.

Major comments:

The title of the paper should refer to a whey protein supplement, rather than “extra protein”.

The authors use “caloric” or calorie” throughout the manuscript where calorie is a measure of energy. Therefore, the authors should be mentioning a VLED or energy intake or energy restricted.

In the title, and throughout, the authors refer to “overweight individuals” which is incorrect and should be “individuals with overweight”

The tables do not explain what the square bracket data is? Is it a range or a 95% CI? Tables 2 and 3 are missing titles of Pre-intervention or baseline, and Change to post intervention. While this can be figured out, it should be explicit, as should the titles for the tables as well. Currently the titles do not describe the table and require the reader to look through the methods to figure out what was done.

When discussing the effects of the physical activity, in the discussion, the authors should refer to resistance training being optimal for maximising gains in FFM.

The authors use muscle mass and FFM interchangeably, where these are not the same and should not be confused in this manner. See lines

The statistical analyses need to be revised. As the study was a parallel groups design, covariates should be used to account for any variation between groups.

Why is Figure 4 not included in the manuscript itself? And why do the authors believe that there were several constipated participants in the CON group? Did the protein assist in not inducing constipation with the low amount of fibre in the diet?

There is no detail included in the manuscript of the adherence of the participants to the VLED. I would be surprised if they were 100% compliant considering real life.

Minor comments

Line 19: change “an extra protein” to “a whey protein”

Line 21: change to “24 h urine collections (Day 0, 7 and 28) to assess nitrogen balance.”

Line 24: change to “no significant group differences”

Line 25: is the FFM change here the change between groups? Or the change between the pre vs post intervention for both groups? Shouldn’t the change here be negative?

Line 32: what does “used for a few weeks” mean? Can the authors be more specific here

Line 40: energy, not calorie

Line 52: “effects of a whey protein supplement before”

Line 54: please add “compared to no additional protein.” Or the like.

Lines 56-58. Change to “The control group followed typical VLED guidelines with no additional energy intake.”

Line 63: change to “women and men, respectively, were”

Line 67: please provide details for the VLED powder provided i.e. brand, company, country etc

Line 69: change to “Test personnel who performed all the physical tests and DXA scans were blinded to the group allocation.”

Line 70: change was to were.

Line 71: please detail the inclusion/exclusion criteria.

Line 72: please detail what the “weekly declaration of adherence” was? Did partipcants have to return any supplements not ingested?

Line 79: “Subjects with overweight (body mass index (BMI) > 28 kg/m2) and aged between 18-55 years were recruited…” and accordingly change line 81 to the exclusion criteria.

Line 83: is the exclusion of diabetics type 1 or 2? Or both?

Line 86-88: this is a result not a methodology.

Line 95: please provide the estimated energy requirements for the participants, to be able to estimate how much energy restriction the participants were in?

Line 100: “The PRO group ingested”

Line 108: change “They” to “Subjects”

Line 110: was the questionnaire standaridsed? Change to “of aerobic and resistance training”

Line 111: lean body mass should be FFM

Lines 111 and 114: “dieting” should be “energy restriction”

Line 113-114: what does “elucidating the maximal muscle perceiving effort” mean? Please reword.

Line 118: change from “at a” to “using a”

Line 121: which version of the enCORE software was used?

Lines 125, 135 (x2) please ensure there is a space between the number and the degrees Celcius symbols.

Line 141: put the macro nutrients in brackets, i.e. “containing 220 kcal (14 g protein, 22 g carbohydrate and 7 g fat).”

Line 150: the workload statement is a result

Line 155: what magnitude of decline was estimated to occur between groups?

Line 161: what covariates were included in the model?

Line 162: what type of regression was used?

Line 167: what does “bad adherence” mean? As in the major comments, the authors need to describe how they measured adherence and what was considered non-compliant (which would be a better term to use than bad adherence!). Please refer here to Figure 4 so the readers can ascertain which groups the drop outs were from.

Line 181: should this also say “no change in absolute or relative weight loss”

Line 196: “greater than the reduction in gynoid fat in both”

Line 200: “After 7 days of the intervention, the NB was different between groups (CON….”

Figure 3. Is the SD missing from the CON bar of Figure 3A?

Line 234: (interaction, p=0.18).

Line 244: “Results from the Aastrand…”

Line 247: check the units here, also ml should be mL

Line 250: change to “comparable reductions in body weight, improvements in body composition and reductions in metabolic risk factors as the individuals that only ingested a VLED.”

Lines 253-254: remove the results here, and change (line 254) to “were comparable with previous…”

Line 258: change to “than PRO, which would also support an increased energy deficit…”

Line 261: “Previously, a protein-rich…”

Line 266: What does “their weight loss due to water excretion during a hypocaloric protein-rich diet has been reported…” – could this be reworded please.

Line 273: add a comma after supplementation

Line 278: add the data for the absolute FFM loss, as the numerical relative numbers are provided.

Section 4.2 or 4.3: add a comment regarding if the exercise had been resistance based (i.e. gold standard for maintenance of FFM)

Line 284: change enhance to “enhances”

Line 285: add a comma after the references on this line

Lines 289-290: at the end of this sentence, add “which achieved 1.08 g/kg/day protein intake overall compared to x.xx g/kg/day in the CON group.”

Lines 292-293: will then change to “Therefore, we cannot exclude that increasing the amount of protein”

Line 295: What is “a markedly weight loss”?

Line 299: what is illuminate? Do the authors mean highlight? How does the intervention do this?

Line 300: fullstop missing here.

Line 301: The Ballor investigation, how long was it for and what was the magnitude of ER? Was it comparable to this investigation?

Line 309: change “actually experience a gain in leg muscle mass” to “gained leg FFM”

Line 316: subscript the 2 in VO2max, and ml to mL

Line 317: “both parameters”, which parameters?

Line 325: change to “ATP production”

Lines 332-336: what about females?

Conclusion section: also highlight that the additional calories did not reduce the weight loss achieved.

Author Response

See the point-to-point response in the uploaded word file 
